# Targeting Mechanisms of the DNA Damage Response (DDR) and DNA Repair by Natural Compounds to Improve cAT-Triggered Tumor Cell Death

**DOI:** 10.3390/molecules27113567

**Published:** 2022-06-01

**Authors:** Jana Aengenvoort, Marlena Sekeres, Peter Proksch, Gerhard Fritz

**Affiliations:** 1Institute of Toxicology, Medical Faculty, Heinrich-Heine University Düsseldorf, Moorenstrasse 5, 40225 Düsseldorf, Germany; jana.aengenvoort@uni-duesseldorf.de (J.A.); marlena.sekeres@uni-duesseldorf.de (M.S.); 2Institute of Pharmaceutical Biology and Biotechnology, Heinrich-Heine University Düsseldorf, Universitätsstrasse 1, 40225 Düsseldorf, Germany; peter.proksch@uni-duesseldorf.de

**Keywords:** DNA damage response, DNA repair, natural compounds, cell death, drug resistance, anticancer drugs

## Abstract

Recently, we identified secalonic acid F (SA), 5-epi-nakijiquinone Q (NQ) and 5-epi-ilimaquinone (IQ) as natural compounds (NC) affecting mechanisms of the DNA damage response (DDR). Here, we further characterized their effects on DDR, DNA repair and cytotoxicity if used in mono- and co-treatment with conventional anticancer therapeutics (cAT) (cisplatin (Cis), doxorubicin (Doxo)) in vitro. All three NC influence the phosphorylation level of selected DDR-related factors (i.e., pCHK1, pKAP1, pP53, pRPA32) in mono- and/or co-treatment. Both SA and NQ attenuate the Cis- and Doxo-induced G2/M-phase arrest and effectively stimulate caspase-mediated apoptosis. Notably, SA impacts DNA repair as reflected by enhanced steady-state levels of Cis-(1,2-GpG)-DNA adducts and Doxo-induced DNA double-strand breaks (DSB). Moreover, SA decreased the mRNA and protein expression of the homologous recombination (HR)-related DSB repair factors RAD51 and BRCA1. Both SA and NQ promote Cis- and Doxo-induced cytotoxicity in an additive to synergistic manner (CI ≤ 1.0). Summarizing, we conclude that SA promotes cAT-driven caspase-dependent cell death by interfering with DSB repair and DDR-related checkpoint control mechanisms. Hence, SA is considered as the most promising lead compound to evaluate its therapeutic window in forthcoming pre-clinical in vivo studies.

## 1. Introduction

Conventional (i.e., genotoxic) anticancer therapeutics (cAT) cause DNA damage by generating DNA adducts, impairing DNA synthesis or by interfering with mitosis, thereby eventually triggering cell death. cAT-induced DNA damage activates a complex stress response program termed the DNA damage response (DDR) [1], which is mainly coordinated by the PI3-like kinase Ataxia telangiectasia mutated (ATM) and the ATM and Rad3-related kinase (ATR) [1,2,3]. While ATM is of major relevance for the regulation of DNA double-strand breaks (DSB)-induced stress responses, ATR gets mainly activated by DNA replication blocking lesions that are causing replicative stress [4,5,6]. The highly complex ATM/ATR-regulated network coordinates cell cycle checkpoint control mechanisms and fine tunes DNA repair and cell death pathways. Hence, the DDR defines the balance between survival and death on the molecular level [7], with p53 being discussed as a key switch between survival and death pathways [8,9,10,11]. Keeping in mind that the DDR is a key player in biology and disease [12] and ATM-/ATR-regulated signaling contributes to tumor cell resistance [13], pharmacological modulation of DNA repair and DDR by DNA repair inhibitors or checkpoint kinase inhibitors is considered as a promising strategy to improve anticancer therapy [14,15,16,17].

Natural compounds (NC) are an important source of anticancer drugs [18,19,20,21,22,23,24,25]. For instance, anthracyclines, taxanes and vinca alkaloids are widely used in anticancer therapy because they inhibit topoisomerase II [26] or affect microtubular function [27], respectively. Apart from inducing tumor cell death on their own, NC can also promote cell death if used in combination with cAT by interfering with proliferation- and death-signaling-related mechanisms [28,29,30,31,32]. For instance, artesunate (Ar), a derivative of the natural compound artemisinin that is used in traditional Chinese medicine (TCM) [33], synergistically increases the sensitivity of ovarian cancer cells to cisplatin (Cis) by downregulating RAD51 and, in consequence, impairing DSB repair [34]. In addition, Ar also activates the DDR, modulates DNA repair [33,35] and promotes the anticancer efficacy of temozolomide in glioblastoma cells by inhibition of homologous recombination repair (HR) [36]. Resveratrol, which is another well-studied NC, also blocks DSB repair [37], causes DNA damage and activates the DDR [38]. Thus, by interfering with DDR-related mechanisms, NCs have a noticeable potential to improve the efficacy of cAT and to overcome inherent and acquired tumor cell resistance, either by promoting DDR-related pro-death pathways and/or by inhibiting DDR-related pro-survival pathways. Overall, NC represent a promising source of novel compounds that can modify DDR and DNA repair mechanisms, thereby triggering cell death of malignant cells on their own and/or improving the efficacy of combination therapy [39,40,41]. Having this in mind, we recently employed a natural compound library containing substances that have been isolated from endophytic fungi, lichens, marine sponges or plants in order to identify novel NC-based bioactive substances, which activate mechanisms of DDR and/or modify DDR-related stress responses evoked by cAT [42]. In this previous study, we identified secalonic acid F (SA), 5-epi-nakijiquinone Q (NQ) and 5-epi-ilimaquinone (IQ) as most promising NC-based anticancer drug candidates affecting the DDR [42], very likely independent of cAT transport. In extension to these studies, we now analyze the influence of SA, NQ and IQ on the activation status of DDR-related factors, DNA repair and cell death induction of malignant cells in more detail. Identical to our previous study [42], we also employed BxPC3 (expressing wild-type Ki-Ras) and SU.86.86 (expressing mutated Ki-Ras) pancreatic carcinoma cells.

## 2. Results

### 2.1. Influence of NC on Mechanisms of the DDR in Pancreatic Carcinoma Cells

In a first set of experiments, we investigated the influence of SA, NQ and IQ on mechanisms of the DDR, which gets activated upon DNA damage formation, notably DSB- and replication-blocking lesions, thereby defining the balance between survival and death [1,7,43,44]. To monitor the activation status of prototypical DDR-related factors, concentrations of the NC were applied that are in the range of their corresponding IC_50_ [42]. In response to mono-treatment with the NC, the most profound activation of the DDR was observed for NQ, which causes a clear increase in the levels of phosphorylated RPA32, ATM, KAP1 and P53 in both BxPC3 and SU.86.86 pancreatic carcinoma cells as compared to untreated control (Figure 1). Following co-treatment with Cis, the protein levels of activated (i.e., phosphorylated) pRPA32 and pKAP1 were further enhanced by ≥2–23-fold as compared to the corresponding mono-treatments while the protein level of pCHK1 was reduced by 70–90% (Figure 1). As opposed to NQ, co-treatment with the chemically related compound IQ had only minor effects on the Cis-mediated stimulation of the DDR (Figure 1). Apart from clearly increasing the protein level of γH2AX as compared to untreated control, mono-treatment with SA did not reveal major alterations in the levels of activated DDR-related factors investigated here. However, if used in combination with Cis, SA reduced the protein levels of pCHK1, pCHK2 and pP53 by about 50–90% as compared to the Cis mono-treatment (Figure 1). Regarding the topoisomerase II poison doxorubicin (Doxo), we observed 50–90% lower protein levels of pChk1, pChk2 and pP53 protein following co-treatment with SA and ≥4-fold increased levels of pCHK1 and pRPA32 after co-treatment of BxPC3 and/or SU.86.86 cells with IQ and NQ, respectively (Figure 1). Protein level of pCHK1 was reduced by about 80% if NQ co-treatment was performed in BxPC3 cells (Figure 1). Taken together, the data show noticeable cAT- and cell-line-related differences in the DDR-modifying potency of the various NC (Appendix A). Based on the data, we suggest that especially SA and NQ impact the cAT-driven activation of checkpoint control mechanisms and replicative stress responses as reflected by altered protein levels of some activated DDR factors (i.e., pCHK1, pCHK2, pP53 and pRPA32).

### 2.2. Cell Cycle Progression and Cell Death Induction Is Differently Affected by NC

Next, we analyzed cell cycle distribution by a flow-cytometry-based method. As measured after a treatment period of 72 h, mono-treatment with SA and NQ triggered apoptosis in BxPC3 cells as indicated by an increase in the percentage of cells present in the subG1 fraction (Figure 2A,B). In addition, both NC further increased cell death following Cis treatment (Figure 2A,B). By contrast, NQ reduced the number of subG1 phase cells following Doxo treatment of BxPC3 cells (Figure 2A,B), indicating that this NC acts in a cAT-specific manner. IQ neither affected Cis- nor Doxo-stimulated cell death (Figure 2A,B). Moreover, both SA and NQ reduced the number of BxPC3 cells present in the G2/M-phase after both Cis and Doxo treatment (Figure 2A,B). At earlier time point of analysis (i.e., 24 h), only SA slightly increased the percentage of cells in subG1 fraction (Figure 2C,D). Moreover, all three NC reduced the frequency of G2/M phase cells following Doxo treatment (Figure 2C,D). Similar effects were observed with SU.86.86 cells (Appendix A). Taken together, the data show that out of the three NC under investigation, SA has the most prominent effects on the activation of cell cycle checkpoint control mechanisms and the induction of cell death in pancreatic carcinoma cells.

### 2.3. Influence of NC on the Activation of Caspase-Related Death Mechanisms

To further clarify whether the NC under investigation activate caspase-dependent apoptotic mechanisms, the cleavage-mediated activation of pro-caspase-3 and pro-caspase-7 was monitored by Western blot analysis. The data show that mono-treatment with SA for 24 h or 72 h causes a clear increase in the protein levels of cleaved (i.e., activated) caspase-7 and, to a lower extent, caspase-3 (Figure 3A,C and Appendix A). A minor increase in the level of cleaved caspase-7 was also observed following 72 h treatment with NQ, while IQ revealed no effects (Figure 3A,C and Appendix A). If combined with cAT, the Cis-stimulated level of cleaved caspase-7 was especially enhanced by SA as analyzed 72 h after drug administration (Figure 3A,C and Appendix A). In addition, caspase-3/7 activity was determined 24 h and 72 h after treatment. The data obtained show that caspase-3/7 activity is significantly stimulated by mono-treatment with SA only (Figure 3B,D and Appendix A), which is in line with the Western-blot-based data. Under situation of combined treatment with Cis, SA and NQ further increased the activity of caspase-3/7 as measured after 24 h and 72 h treatment, respectively (Figure 3B,D and Appendix A). In combination with Doxo, SA promoted caspase-3/7 activity after 24 h while reducing it at a later time point (i.e., 72 h). Taken together, the results show that both SA and an NQ influence caspase-3/7-related mechanisms of apoptosis in a cAT- and time-dependent manner.

### 2.4. Impact of NC on the Formation of CisPt-Induced DNA Intrastrand Crosslinks

To monitor the influence of the NC on DNA damage formation, the steady-state level of Cis-induced DNA intrastrand crosslinks (ICL) was analyzed after 24 h treatment period with Cis by Southwestern blot analysis. The data obtained show a significant increase in the level of Cis-induced ICL in SU.86.86 cells co-treated with SA (Figure 4B). A tendential increase in Cis-induced ICL by SA was also observed in BxPC3 cells (Figure 4A) and, moreover, following co-treatment with NQ but not with IQ (Figure 4A,B). Hence, especially SA increases the steady-state level of ICL formed by Cis. Based on these data we suggest that SA, and to a lower extent NQ, impairs the repair of Cis-induced ICL or affects mechanisms of Cis transport.

### 2.5. Formation and Repair of cAT-Induced DSB Is Affected by NC

Next, we investigated if NC influence the repair of cAT-induced DNA damage in more detail. To this end, the initial formation and the residual number of nuclear γH2AX foci, which is a surrogate marker of DNA double-strand breaks (DSB), was monitored. To induce DSB, the cells were pulse-treated for 4 h with either Cis or Doxo. To enable DNA repair, cells were post-incubated for another 24 h in drug-free medium or in medium containing the NC before the number of remaining nuclear γH2AX foci was monitored (Figure 5A). Of note, mono-treatment with all NC slightly increased the number of nuclear γH2AX foci as compared to the untreated control as anticipated (Figure 5B).

Regarding Cis, we found that the number of nuclear γH2AX foci was low immediately after the pulse-treatment period and further increased within the post-incubation period of 24 h (Figure 5B), indicating that Cis leads to a delayed formation of DSB. Exposure to SA, NQ or IQ did not increase the mean number of residual Cis-induced nuclear γH2AX foci (Figure 5B). However, almost 50% of the Cis-treated cells that were post-incubated with SA revealed nuclear pan-staining (Appendix A), indicating that extensive DNA damage is caused by Cis over time in the presence of SA. Previously, γH2AX pan-staining was suggested to result from nucleotide excision repair (NER) [45,46], which is a major DNA repair pathway also for Cis-induced ICL. γH2AX pan-staining following Cis exposure was not observed in the presence of NQ and IQ (Appendix A), demonstrating that this effect is specific for SA. In case of Doxo mono-treatment, we observed a higher number of nuclear γH2AX foci immediately after the pulse-treatment, which decreased over the 24 h post-treatment period, reflecting the repair of DSB. SA and NQ significantly enhanced the number of residual nuclear γH2AX foci resulting from the Doxo pulse-treatment, while IQ did not show this effect (Figure 5C). The data are indicative of a reduced repair of Doxo-induced DSB by SA and NQ. Based on the data, we suggest that SA inhibits the repair of both Doxo-induced DSB and Cis-induced ICL, leading to a higher number of residual DSB and nuclear γH2AX pan-staining, respectively.

### 2.6. Impact of NC on the mRNA and Protein Expression of Susceptibility-Related Factors

To characterize the influence of the NC under investigation on the mRNA expression of a selected subset of susceptibility-related factors, quantitative RT-PCR analyses were performed 24 h after compound addition using a 96-well plate-based array. At the doses used for these mRNA expression analyses, all NC tested are cytotoxic in pancreatic carcinoma cell lines and are able to activate the DDR on their own [42]. The genes selected for analysis were classified into factors involved in DDR and DNA repair (e.g., BRCA1, Chk1/2, MSH2, RAD51, XRCC3), apoptosis (e.g., BAX, BCL-2, FASL, FASR), oxidative stress (e.g., GPX1, HMOX1), stress response in general (e.g., HSP90) and drug transport (e.g., ATP7A, MRP2) (Figure 6A). SA and NQ mono-treatment caused the most substantial (i.e., ≥2.0-fold or ≤0.5-fold) alterations in the mRNA expression of some of the aforementioned genes, whereas IQ had only minor effects (Figure 6A). In more detail, SA especially stimulated the mRNA expression of GADD54a, FASL and HSPA1B while largely reducing the mRNA levels of BRCA1, BRCA2, CHK1, CHK2, MRE11, RAD51 and FASR (Figure 6A). NQ caused upregulation of mRNA levels of GADD45a, BCL2, CASPASE 9, DYRK1VB, HMOX1, MTA1 and ATP7A while causing a noticeable suppression of FASL mRNA level (Figure 6A). IQ only caused a 2-fold upregulation of ATP7A mRNA expression and a suppression of FASL (Figure 6A). Data obtained from the RT-qPCR analyses are summarized in Appendix A, highlighting that SA predominantly alters the mRNA expression levels of genes involved in the regulation of the DDR and DNA-repair, while NQ majorly affect the mRNA expression of factors related to anti-oxidative stress response and apoptosis.

Based on the mRNA expression results, protein levels of selected factors were additionally determined by Western blot analyses. In line with the results obtained from the mRNA analyses, protein expression of RAD51 and BRCA1, which are involved in DSB repair by homologous recombination (HR), was also found to be substantially downregulated (>50%) following SA treatment (Figure 6B). Moreover, also in line with the mRNA data, SA causes a moderate decrease in the protein expression of MRE11, CHK1 and CHK2 by about 40% and that of the anti-apoptotic factor BCL-2 by about 80% (Figure 6B,C). The latter result is in line with the observation that SA stimulated apoptosis as reflected by a significant increase in the percentage of cells present in the subG1 fraction and by the observed stimulation of caspase 3/7 activity (see Figure 2 and Figure 3). The protein levels of FASL, FASR, BAX and GADD45a remained largely unchanged by SA at the time point of analysis (Figure 6B,6C) and, hence, do not reflect the alterations observed in mRNA levels. This might be due to different time kinetics of mRNA and protein expression following SA treatment or variations in mRNA and protein stability.

### 2.7. Additive and Synergistic Cytotoxic Effects of NC

To investigate possible additive or synergistic cytotoxic effects if cAT is combined with NC, cell viability was analyzed and combination index (CI) was determined [47,48]. Summarizing the results of these extensive analyses (for detailed data please see Appendix A) synergistic effects were observed for Cis and NQ in both pancreatic carcinoma cell lines (Figure 7A,C). In combination with Doxo, NQ mediates additive toxicity (Figure 7B,D). SA caused additive toxicity if combined with Cis or Doxo, except in BxPC3 cells, where weak synergism was found with Cis (Figure 7). In most cases, additive toxicity was also found if IQ was used for co-treatment, except regarding Doxo where IQ caused a slight synergism in BxPC3 cells (Figure 7).

### 2.8. Cytotoxic Activity of NC on Various Tumor Cell Lines

Finally, we comparatively investigated the susceptibility of various human tumor cell lines to a mono-treatment with the NC under investigation. The data obtained show similar IC_50_ (i.e., 3–5 µM) for SA in pancreatic carcinoma cells (BxPC3, Su.86.86), colon carcinoma cells (HCT116, HT29) and lung cancer cells (H1975), whereas triple negative mammary carcinoma cells (MDA-MB-231) revealed an approximately 10-fold higher IC_50_ (i.e., 35 µM) (Appendix A). Regarding NQ, it was HT29 cells that were most resistant to this compound as evidenced by the highest IC_50_ (i.e., 20 µM) as compared to the other tumor cell lines test (IC_50_: 2–10 µM) (Appendix A). Colon and lung cancer cells were more resistant to IQ (i.e., IC_50_: 20–40 µM) as compared to the pancreatic carcinoma cells (IC_50_: 1.5–8 µM) (Appendix A). Taken together, the cytotoxic potency of SA, NQ and IQ is cell line-specific with SA evoking cytotoxicity in the low micromolar range in the majority of the tumor cell lines tested. Of note, non-malignant primary human fibroblast cells, which were included for control, were more resistant to SA and IQ (IC_50_: SA, 40 µM; NQ, 5 µM; IQ, 60 µM) (Appendix A) as compared to the malignant cells.

## 3. Discussion

### 3.1. Influence of NQ, IQ and SA on Mechanisms of the DDR

The PI3-like kinases ATM and ATR play a key role in the regulation of the DDR that is triggered upon the formation of DNA damage, notably DSB and replication stress [1,2], thereby defining the balance between survival and death [7]. Investigating the impact of the pre-selected NC (i.e., SA, NQ, IQ) on the protein level of a subset of DDR-related factors under situation of mono- and combined treatment with cAT, substantial NC-specific differences were observed (see Figure 1 and Appendix A). As concluded from the results obtained by the mono-treatment, 5-epi-nakijiquinone Q (NQ) causes the strongest activation of the DDR as reflected by an increase in the protein levels of pRPA32, pATM, pKAP1 and pP53 as compared to the untreated control. If used in combination with Cis, the levels of pRPA32 and pKAP1 were higher than those of the corresponding mono-treatments, while the level of pChk1 was reduced. A stimulatory effect of NQ on the activity of DDR-related kinases has not yet been reported so far. Rather, it has been shown that NQ is able to inhibit the kinases Aurora-B and NEK6 [49] that are involved in the regulation of DNA repair [50,51]. Moreover, NQ influences Her-2/Neu receptor tyrosine kinase signaling [52], which affects mechanisms of DNA repair [53]. Hence, our data indicate that NQ does not only inhibit kinases related to the regulation of DNA repair but, moreover, also influences kinases related to the DDR. Of note, the structurally related compound 5-epi-ilimaquinone (IQ) revealed no major effects on the level of phosphorylated DDR-related proteins if used in mono-treatment. If used in combination with Doxo, IQ further increased the protein level of pChk1 in BxPC3 cells only. Noteworthy in this context, IQ has been reported to stabilize p53 protein by promoting Ser15 phosphorylation, and, furthermore, to cause upregulation of p21 protein in colon cancer cells if used at concentrations of ≥5 µM [54]. Similar to IQ, SA did not show major effects on the level of phosphorylated DDR-related proteins if used in mono-treatment. However, if used in combination with both Cis and Doxo, SA caused a noticeable reduction in the protein levels of pCHK1, pCHK2, pP53 and pKAP1 as compared to the corresponding cAT mono-treatment. So, our data demonstrate that SA causes the most profound impairment of the activity of DDR-related factors stimulated by cAT as compared to NQ and IQ. So far, secalonic acid F (SA) was reported to inhibit PI3K/Akt/β-catenin signaling [55], which is noteworthy having in mind that ATM and ATR belong to the family of PI3-like kinases. Summarizing, we presented novel evidence that NQ and SA interfere with mechanisms of the DDR, in particular with replicative-stress-related responses evoked by Cis and Doxo, as reflected by changes the protein levels of pCHK1, pCHK2, pRPA32 and pP53. This finding points to NQ and especially SA as highly attractive lead compounds for a pharmacological modulation of DNA repair- and DDR-related kinases.

### 3.2. Modulation of Cell Cycle Progression and Cell Death Pathways by NC

Analyzing the impact of the NC on cell cycle distribution by flow-cytometry-based methods after a treatment period of 72 h, we observed that mono-treatment with SA and NQ increases the percentage of cells present in the subG1 fraction and, moreover, augments the frequency of cells present in subG1 fraction following Cis treatment. In addition, both NC attenuated the Cis- and Doxo-stimulated increase in the percentage of BxPC3 cells present in the G2/M-phase. At the earlier time point of analysis (i.e., 24 h) after Doxo treatment, all three NC reduced the frequency of G2/M phase cells. This data show that each of the NC impacts the cAT-driven activation of cell cycle checkpoint control mechanisms. In line with the flow-cytometry-based data, SA and NQ revealed the strongest cleavage and activation of executor capases-3 and -7, respectively, indicating that cell death induced by these NC is caspase-3/7-dependent. In line with this, it has been reported that NQ triggers cell death in L5178Y murine lymphoma cells [49] and SA causes apoptosis by downregulation of the anti-apoptotic factors MCL-1/BCL-2 [55]. Of note and in line with this report, we also observed a reduced mRNA and protein expression of BCL-2 following SA treatment in our study. Anticancer activity of SA has been shown in HCC a nude mouse model [55], which demonstrates its preferable pharmacokinetic properties in vivo. Furthermore, SA is more efficient in cell death induction than the antimetabolite 5-FU, both in vivo and in vitro [56], again supporting the hypothesis of the favorable anticancer activity of this compound. Cytotoxic activity of IQ has been related to G1-arrest and S-phase disturbance as well as nuclear translocation of the growth arrest and DNA damage-inducible gene 153 (CHOP/GADD153) [57]. However, substantial alterations in cell cycle progression and induction of apoptotic cell death by IQ were not observed in our study. Taken together, and in line with the DDR-related data discussed before, we hypothesize that it is especially SA and NQ that have the most noticeable impact on the activation of cell cycle checkpoints. Furthermore, out of the three NC under investigation, SA is the strongest activator of mechanisms of cAT-triggered apoptosis and, importantly, has been proven to have anticancer activity in vivo.

Having in mind the manifold types of cell-death-related pathways and their high complexity, we additionally monitored the cytotoxic potency of the three NC in combination with the cAT by determining the combination index (CI). Here, synergistic toxicity was observed if Cis was combined with SA or, even more pronounced, with NQ. If combined with Doxo, both NC revealed additive toxicity. These data support the view that combined treatment with cAT and SA or NQ causes additive to synergistic toxicity in malignant cells. Mono-treatment with SA and NQ also provoked cytotoxicity in colon and lung carcinoma cells already at low micromolar concentration. Hence, SA and NQ appear to be particular useful for forthcoming studies aiming to investigate their efficacy as anticancer therapeutics in vivo.

Since adverse normal tissue damage is a severe problem in oncology, it is important to figure out whether the NC are well tolerated in normal cells. Moreover, it is important to rule out the possibility that NC evoke additive or even synergistic adverse toxicity on healthy tissue if used in a combined treatment regimen. Regarding the response of non-malignant primary human fibroblasts to NC treatment, IC_50_ of 40 µM, 5 µM and 60 µM were found for SA, NQ and IQ, respectively (Appendix A). Thus, NQ reveals similar toxicity in malignant and non-malignant cells speaking against a favorable therapeutic window of this NC. By contrast, the IC_50_ of SA in primary fibroblasts was >10-fold higher as compared to the majority of the tumor cells (Appendix A). Therefore, we hypothesize that SA might be a well-tolerated NC-based substance with a reasonably wide therapeutic window. This hypothesis remains to be scrutinized by forthcoming preclinical in vivo studies.

### 3.3. Inhibitory Effects of SA on Mechanisms of DNA DSB Repair

Having in mind that cAT-induced DNA damage and the activation of related cell death pathways are majorly affected by the DNA repair capacity of cells [58,59], we additionally investigated the influence of the NC on the repair of cAT-induced DNA damage. Analyzing the level of Cis-induced DNA intrastrand crosslinks (ICL) by Southwestern blot analyses, we observed that the level of residual ICL was enhanced following SA treatment. This indicates that the repair of Cis-induced ICL by nucleotide excision repair (NER) might be retarded by SA. In line with this hypothesis we found that the mRNA expression of ERRC1, which is involved in the repair of ICL by NER and has been hypothesized as a predictive factor of Cis response in bladder [60] and lung carcinoma [61], was decreased by SA. In line with this, the protein expression of ERCC1 was also moderately altered at the time point of analysis. Of note, inconsistencies between mRNA and protein levels of ERCC1 are described by others [62], warranting further time kinetic analyses.

Apart from alterations in the mRNA level of ERCC1, RT-qPCR analyses also revealed a large decrease in the mRNA expression of various DNA repair- and DDR-related factors, including BRCA1/2, RAD51, MRE11, MLH1, MSH2, XRCC3 and CHK1/2, by SA. Most important, the attenuated mRNA expression of BRCA1 and RAD51, which are involved in the repair of DNA double-strand breaks by homologous recombination (HR), was confirmed on the protein level. This finding indicates that SA impacts the HR-mediated repair of DSB formed upon cAT treatment by downregulating the mRNA and protein expression of key factors of HR repair. To further address this point, the formation and repair of nuclear γH2AX foci, which are indicative of DSB, was analyzed. Data obtained show that SA significantly slows down the removal of nuclear γH2AX foci that have been generated by pulse-treatment with the topo II poison doxorubicin. Taken together, data obtained from immunohistochemistry-based foci and mRNA expression analysis concomitantly indicate that SA is able to disturb the repair of DSB, very likely by impacting gene expression of factors involved in HR. An interference of SA with DSB repair has not yet been reported so far and, hence, deserves further investigations.

Summarizing, three NC candidate compounds that have been obtained from previous study [42] were further characterized regarding their mode of action in vitro. Data obtained are summarized in a hypothetical model (Figure 8). Based on our data together with published data, we speculate that IQ mainly causes oxidative DNA damage by increasing ROS formation [63,64], while NQ is able to both stimulate and inhibit the activity of DDR-related kinases. We assume that SA can inhibit both replicative stress-related responses that are triggered by cAT treatment and, in addition, HR-related mechanisms of DSB repair by inhibiting the expression of corresponding repair genes. In consequence of impaired DSB repair and G2/M-abrogation caspase 3/7-driven cell death is promoted. Based on the data and having in mind that anticancer activity of SA has already been reported in vivo [55,56], we suggest SA as the most promising NC for forthcoming preclinical studies to investigate its anti-tumor effectiveness in mono- and combined-treatment regimen and to figure out its therapeutic window in vivo.

## 4. Materials and Methods

### 4.1. Materials

The marine-sponge-derived compounds 5-epi-nakijiquinone Q (NQ) and 5-epi-ilimaquinone (IQ) as well as the fungal compound secalonic acid F (SA) were isolated following a screening of a natural compound library containing around 300 natural compounds (NC) from endophytic fungi, lichens, marine sponges or plants (P. Proksch; Institute of Pharmaceutical Biology and Biotechnology of the Heinrich Heine University (Düsseldorf, Germany) [42]. All natural compounds were dissolved in DMSO. Doxorubicin (Doxo) originates from Cellpharm (Bad Vilbel, Germany), cisplatin (Cis) from TEVA Pharmaceutical Industries Limited (Ulm, Germany). Antibodies detecting CHK2, Cis-(1,2-GpG), ERCC1, pATM (S1981), pCHK2 (T68) or RAD51 were purchased from Abcam (Cambridge, UK), pKAP1 (S824) and pRPA32 (S4/S8) from Bethyl Laboratories Inc. (Montgomery, Texas, USA), 53BP1, BCL-2, BRCA1, CHK1, Cleaved Caspase-3, Cleaved Caspase-7 (D198), GADD45A, GAPDH, pATR (S428), pCHK1 (S345), pP53 (S15), PARP, pro-caspase-3 and Talin-1 from Cell Signaling Technology (Cambridge, UK). Ser139-phosphorylated histone 2AX (γH2AX) was purchased from Millipore (Billerica, MA, USA), FASL, p53 and ß-actin specific antibody from Santa Cruz Biotechnology (Santa Cruz, CA, USA) and pH3 (S10) from Thermo Fisher Scientific Inc. (Waltham, MA, USA). The fluorophore-conjugated secondary antibody Alexa Fluor^®^ 488 goat anti-mouse or anti-rabbit IgG were bought from Life Technologies (Carlsbad, CA, USA) and the horseradish peroxidase-conjugated secondary antibody goat anti-mouse, anti-rabbit or anti-rat IgG from Rockland (Rockland, Limerick, PA, USA).

### 4.2. Cell Culture and Drug Treatments

Pancreatic carcinoma cells expressing oncogenic Ki-Ras (SU.86.86) and lung carcinoma cells (NCI-H1975) were obtained from the American Type Culture Collection (ATCC) (Manassas, VA, USA). BxPC3 pancreatic carcinoma cells harboring wild-type Ki-Ras, colon carcinoma cells HCT-116 and HT29, as well as triple negative breast cancer cells (MDA-MB-231) originate from the German Collection of Microorganisms and Cell Culture (DSMZ) (Braunschweig, Germany). BxPC3, SU.86.86, HT29 and NCI-H1975 cells were cultured in RPMI-1640 medium, MDA-MB-231 cells in Dulbecco’s modified Eagle medium and HCT-116 cells were cultured in McCoy’s 5A medium (all media were from Sigma, Steinheim, Germany), each supplemented with 10% heat-inactivated fetal bovine serum (Biochrom, Berlin, Germany) and 1% penicillin/streptomycin (Sigma, Steinheim, Germany) at 37 °C in a humidified atmosphere containing 5% CO_2_. Cells were treated 24 h after seeding.

### 4.3. Determination of Cell Viability

Cell viability was determined using the Alamar Blue^®^ assay [65], which measures the reduction of the non-fluorescent dye resazurin to the fluorescent metabolite resorufin. At a time of 4–72 h after drug treatment, cells were incubated with 44 μM resazurin sodium salt (Sigma, Steinheim, Germany) in Dulbecco’s modified Eagle medium w/o phenol red (Sigma, Steinheim, Germany) for 2.5 h before fluorescence was measured (excitation: 535 nm, emission: 590 nm, 5 flashes, integration time: 20 µs (Tecan infinite 200, Tecan, Männedorf, Switzerland)). Mean fluorescence intensity is proportional to cell viability, which is displayed relative to the respective untreated control (=100%).

### 4.4. Combination Index

To determine synergistic or additive effects of combination treatments on the viability, the rates of cell growth inhibition were obtained from Alamar Blue^®^ assays. The combination index (CI) was calculated using Compusyn software version 1.0 (ComboSyn, Inc., Paramus, NJ, USA) based on the Chou–Talalay method [66]. We considered synergistic, additive or antagonistic effects if CI < 0.8, CI ≥ 0.8 ≤ 1.2 or CI > 1.2, respectively.

### 4.5. Analysis of Nuclear Foci Formation

The formation of nuclear foci resulting from ATM/ATR-catalyzed Ser139 phosphorylation of histone 2AX (γH2AX) was measured as a surrogate marker of DSB by fluorescence microscopy [67,68,69,70]. To this end, cells were seeded onto cover slips, treated with the indicated doses of compounds and were fixed with 4% formaldehyde/PBS (15 min; RT) followed by an incubation with ice-cold methanol (≥20 min; −20 °C). Subsequently, cells were blocked in 5% BSA in 0.3% Triton X-100/PBS (1 h; RT), incubated with primary antibody specifically detecting phosphorylated (Ser139) histone 2AX (dilution 1:500 in 5% BSA in 0.3% Triton X-100/PBS; 16 h; 4 °C) and incubated with fluorophore-labeled secondary antibody (dilution 1:1000 in 5% BSA in 0.3% Triton X-100/PBS; 90 min; RT, in the dark). Cells were mounted in Vectashield containing DAPI (Vector Laboratories, Burlingame, CA, USA) and analyzed with Olympus BX43 microscope (Olympus, Hamburg, Germany). The number of γH2AX foci per nucleus was quantified.

### 4.6. Western Blot Analysis

The phosphorylation status of key factors of the DDR (i.e., p53, CHK1/2, KAP1, RPA32) was monitored by Western blot analyses**.** Total cell extracts were collected in lysis buffer (50 mM Tris–HCl, 150 mM NaCl, 2 mM EDTA, 1% NP-40, 0.1% sodium dodecyl sulfate, 1% sodium desoxycholate, 1 mM sodium orthovanadate, 1 mM phenylmethylsulfonyl fluoride, 50 mM sodium fluoride, 1 × protease inhibitor cocktail (Cell Signaling, Beverly, MA, USA)) at the indicated time point after treatment. After sonication protein concentration was determined by the DC™ Protein Assay (Bio-Rad Laboratories, Hercules, CA, USA). An amount of 30–50 μg of protein was denatured by heating (5 min; 95 °C), separated by SDS-PAGE (6–15% gels) and transferred onto nitrocellulose membranes by wet blotting using Mini-PROTEAN^®^ electrophoresis chamber (Bio-Rad Laboratories, Hercules, CA, USA). In general, membranes were cut into 2–3 strips spanning defined molecular weights before incubation with the appropriate primary antibody was performed. Thereby, and by re-probing the membranes, we aimed to achieve more efficient and meaningful comparative analyses of protein expression patterns. Membranes were blocked in 5% non-fat milk in TBS/0.1% Tween 20 (MERCK (Darmstadt, Germany)) (2 h, RT) and incubated with the corresponding primary antibody (1:500 to 1:10,000; overnight; 4 °C). After washing with TBS/0.1% Tween 20, the secondary (peroxidase-conjugated) antibody was added (1:2000; 2 h; RT). Chemiluminescence detection of protein–antibody complexes was performed with Chemidoc (Bio-Rad Laboratories, Hercules, CA, USA). For quantification, densitometrical analyses were performed using the open-source software ImageJ that was developed by the National Institutes of Health (NIH, Bethesda, Maryland, USA) (https://imagej.nih.gov/ij/).

### 4.7. Analyses of Cell Cycle Progression by Flow Cytometry

Cell distribution was analyzed by flow cytometry. Therefore, cells were trypsinized, washed and resuspended in PBS. Cells were fixed in ice-cold ethanol (≥20 min; −20 °C). After centrifugation (800× *g*; 10 min; 4 °C) the supernatant was discarded, the cells were resuspended in PBS containing RNase A (Serva Electrophoresis GmbH (Heidelberg, Germany)) (1 μg/μL) and incubated for 1 h at room temperature. Next, propidium iodide (Sigma-Aldrich, St. Louis, MO, USA) was added and the cells were subjected to flow cytometric analysis. BD Accuri™ C6 Flow Cytometer (BD Biosciences (San Jose, CA, USA)) was used for quantification of the percentage of cells present in SubG1-fraction and G2/M-phase of the cell cycle. SubG1-fraction was considered as a measure of dead cells.

### 4.8. Determination of Apoptotic Cell Death

Activation of effector caspases 3 and 7 was examined using the Apo-ONE^®^ homogeneous Caspase-3/7 Assay Kit (Promega, Madison, Wisconsin, USA) which was used according to the manufacturer’s protocol. In this assay, the non-fluorescent caspase 3 and 7 substrate rhodamine 110 (bis-(NCBZ- L-aspartyl-L-glutamyl-L-valyl-L-apartic acid amide; Z-DEVD-R110) is sequentially cleaved by caspase 3 and 7, releasing the fluorescent dye rhodamine 110. An amount of 100 μL of the Apo-ONE^®^ substrate was added to 100 μL residual medium and the fluorescence was measured after 1 h at room temperature in the absence of light (excitation: 499 nm; emission maximum: 521 nm). The fluorescence reflects the activity of the caspases 3 and 7 and was normalized to untreated control.

### 4.9. Analysis of Platinum-Induced DNA Intrastrand Crosslinks

With Southwestern blot analyses, the level of Cis-(1,2-GpG) intrastrand crosslinks was investigated. To this end, genomic DNA was isolated using the DNeasy Blood and Tissue kit (Qiagen (Hilden, Germany)). Concentration and purity of the DNA was measured photometrically with the NanoVue™ Plus Spectrophotometer (GE Healthcare (Little Chalfont, UK)). An amount of 1 µg of the DNA was diluted in TE buffer, denatured by heating (10 min, 95 °C), cooled on ice and 100 μL ice cold ammonium acetate (2 M) was added. Next, the DNA was transferred onto a nitrocellulose membrane that was soaked in 1 M ammonium acetate by use of a slot-blot apparatus (Carl Roth GmbH (Karlsruhe, Germany)) and a vacuum pump. After washing the membrane with 1 M ammonium acetate and water, it was incubated with 5 × SSC (10 × SSC: 1.5 M NaCl, 150 mM sodium citrate, pH 7.0) for 5 min and baked for 2 h at 80 °C before it was blocked in 5% non-fat milk in TBS/0.1% Tween 20 (1 h; RT). Following washing steps incubation with the primary antibody directed against Cis-(1,2-GpG) intrastrand crosslinks (1:5000) (Liedert et al., 2006) was performed (16 h; 4 °C), followed by another washing step and the subsequent incubation with peroxidase conjugated anti-rat IgG secondary antibody (1:5000; 2 h; RT). Visualization of the Cis-(1,2-GpG) intrastrand crosslinks was performed by chemiluminescence (BM Chemiluminescence Blotting Substrate (POD) (Hoffman-La Roche, Basel, Schweiz)) with Chemidoc (Bio-Rad Laboratories, Hercules, CA, USA). Additionally, the membrane was stained with methylene blue to ensure equal DNA loading. Autoradiographies were densitometrically analyzed with Image Lab™ Software Version 6.0.1. (Bio-Rad, Hercules, California, USA).

### 4.10. Quantitative Real-Time PCR-Based mRNA Expression Analyses

Putative markers of Cis susceptibility were selected on the basis of a review by Galluzzi et al. [71] who has classified putative Cis resistance factors of tumor cells. Based on this report we assembled a 96-well-based quantitative real-time (qRT) PCR array to analyze the mRNA expression of these factors [72]. Total RNA was purified using the RNeasy Mini Kit (Qiagen, Hilden, Germany). The reverse transcriptase (RT) reaction was performed with the High Capacity cDNA Reverse Transcription Kit (Applied Biosystems, Darmstadt, Germany) using 2000 ng of RNA. For each PCR reaction, 20 ng of cDNA and 0.25 μM of the corresponding primers (Eurofins MWG Synthesis GmbH, Ebersberg, Germany) were used. Quantitative RT-polymerase chain reaction (PCR) was performed as follows: 1. 95 °C—10 min; 2. 35-40 amplification cycles with 95 °C—15 s, 55 °C—15 s, and 72 °C—17 s; 3. 95 °C—1 min, 55 °C—1 min, 65 °C—5 s. Analyses were performed in triplicates using a CFX96 cycler (BioRad) and the SensiMix SYBR Kit (Bioline, London, UK). The primers used for mRNA expression analyses are listed in Appendix A. At the end of the run, melting curves were analyzed to ensure the specificity of the amplification product. mRNA expression levels were normalized to those of β-actin and GAPDH. If not stated otherwise, relative mRNA expression of untreated control cells was set to 1.0.

### 4.11. Statistical Analysis

If not stated otherwise, the unpaired, two-tailed Student’s *t*-test (marked with *) and one-way ANOVA (marked with #) were used for statistical analyses. *p*-values *p* ≤ 0.05 were considered as statistically significant.

## Figures and Tables

**Figure 1 molecules-27-03567-f001:**
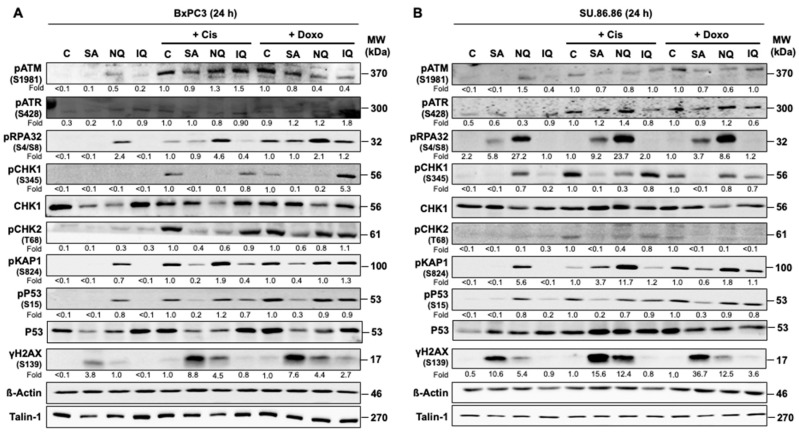
Effects of NC on mechanisms of the DDR. Western blot analyses were performed in order to examine the activation status of selected factors of the DNA damage response (DDR) after treatment with the natural compounds (NC) either as mono-treatment or combination treatment with cAT. To this end, BxPC3 (**A**) and SU.86.86 (**B**) cells were treated for 24 h with the previously identified natural compounds (NC) [42] (SA, 3 µM; NQ, 5 µM; IQ, 1.5 µM). For the combination treatment, the cells were additionally treated with cisplatin (5 µM) (Cis) or doxorubicin (0.3 µM) (Doxo). After treatment, total cell extracts were collected for Western blot analysis. Expression of β−actin and talin-1 were used as loading controls. Shown is the result of a representative experiment. Relative expression levels of phosphorylated (i.e., activated) DDR-related proteins in the respective Cis- or Doxo-treated cells were set to 1.0. For quantification of the NC mono-treated cells, the Cis-treated cells were taken for control and set to 1.0. Only differences in protein expression of >1.5-fold and <0.7-fold were considered as biologically noticeable. ATM, ataxia telangiectasia mutated; ATR, ataxia telangiectasia and RAD3 related; C, Control; CHK, checkpoint kinase; H2AX, histone 2AX; IQ, 5-epi-ilimanquinone; KAP1, KRAB-associated protein 1; NQ, 5-epi-nakijiquinone Q; p, phospho-; P53, tumor suppressor gene; RPA32, replication protein A2, 32 kDA subunit; SA, secalonic acid F; γH2AX, S139 phosphorylated H2AX.

**Figure 2 molecules-27-03567-f002:**
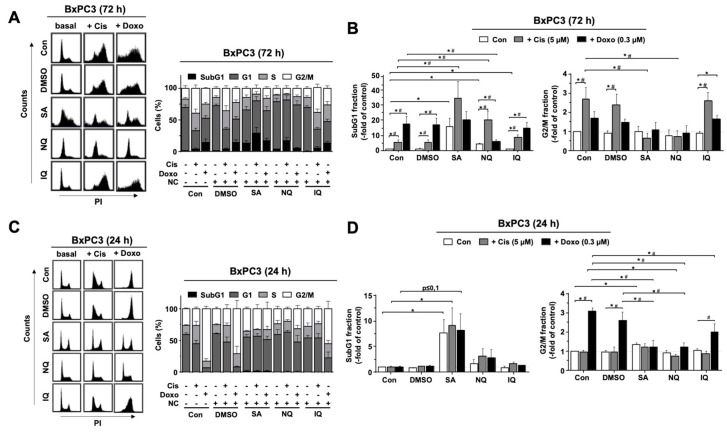
Effect of NC on cell cycle progression. Logarithmically growing pancreatic carcinoma cells (BxPC3) were treated with the various NC (SA, 3 µM; NQ, 5 µM; IQ, 1.5 µM). For the co-treatment with cAT, cells were additionally exposed to cisplatin (5 µM) (Cis) or doxorubicin (0.3 µM) (Doxo). The percentage of cells present in different phases of the cell cycle was measured 24 h and 72 h later by flow cytometry. (**A**,**C**) Representative flow-cytometry-based data and percentage of cells in different cell cycle as compared to non-treated control cells as analyzed 72 h (**A**) or 24 h (**C**) after drug administration. Shown are mean values + SEM obtained from three independent experiments. (**B**,**D**) Cells in SubG1 fraction or G2/M phase of the cell cycle as compared to non-treated control cells. Shown are mean values + SEM obtained from three independent experiments. *^,#^
*p* ≤ 0.05. Con, control; DMSO, dimethyl sulfoxide; IQ, 5-epi-ilimanquinone; NC, natural compound; NQ, 5-epi-nakijiquinone Q; PI, propidium iodide; SA, secalonic acid F.

**Figure 3 molecules-27-03567-f003:**
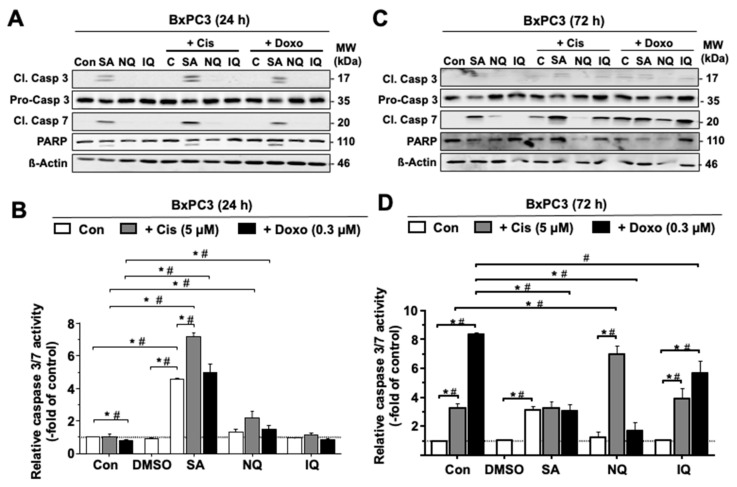
Activation of caspase-3 and -7 by NC. BxPC3 pancreatic carcinoma cells were treated with natural compounds only (SA, 3 µM; NQ, 5 µM; IQ, 1.5 µM) or in combination with cisplatin (Cis) or doxorubicin (Doxo) for 24 h (**A**,**B**) or 72 h (**C**,**D**). (**A**,**C**) Protein level of cleaved pro-caspase-3 (Cl. Casp 3) and -7 (Cl. Casp 7) was analyzed by Western blot analysis. (**B**,**D**) Caspase-3/7 activity was examined by use of the Apo-ONE^®^ assay as described in methods. Quantitative data shown in (**B**,**D**) are the mean + SEM from three independent experiments each performed in triplicate. *^,#^
*p* ≤ 0.05. Con, control; IQ, 5-epi-ilimanquinone; NQ, 5-epi-nakijiquinone Q; SA, secalonic acid F. Data obtained with SU.86.86 cells are shown in Appendix A.

**Figure 4 molecules-27-03567-f004:**
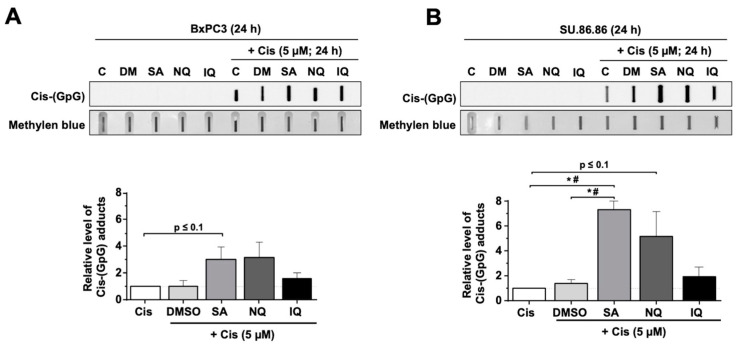
Influence of NC on the formation of DNA intrastrand crosslinks (ICL). Pancreatic carcinoma cell lines, (**A**) BxPC3 and (**B**) SU.86.86, were incubated for 24 h with the natural compound (SA, 3 µM; NQ, 5 µM; IQ, 1.5 µM) in the presence or absence of cisplatin (Cis). The steady-state amount of (GpG) intrastrand crosslinks was determined by Southwestern blot analysis using anti-(GpG)-specific antibody as described in methods. For quantification, the blots were analyzed densitometrically. Shown are the mean + SEM from *n* = 3 independent experiments. *^,#^
*p* ≤ 0.05. C, control; DMSO, dimethyl sulfoxide; IQ, 5-epi-ilimanquinone; NQ, 5-epi-nakijiquinone Q; SA, secalonic acid F.

**Figure 5 molecules-27-03567-f005:**
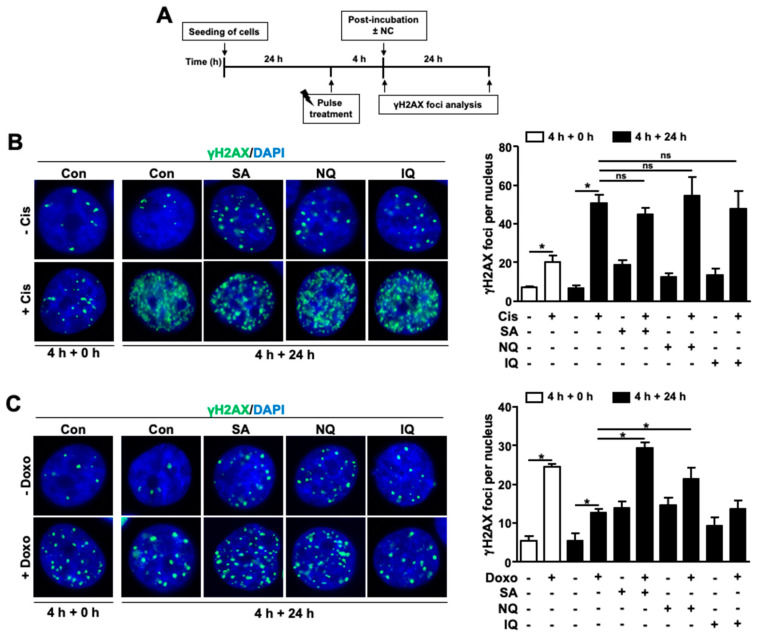
Impact of NC on the repair of cAT-induced DSB. (**A**) Schematical illustration of the treatment scheme. (**B**,**C**) In order to determine if the NC affect the repair of cAT-induced DSB, BxPC3 pancreatic carcinoma cells were pulse-treated (4 h) with Cis (20 µM) (**B**) or Doxo (0.5 µM) (**C**) followed by a post-incubation period (24 h) in drug-free medium or in the presence of the NC (SA: 3 µM, NQ: 2 µM, IQ: 8 µM). Afterwards, the number of residual nuclear γH2AX foci was analyzed as a surrogate marker of DSB by immunofluorescence staining. Shown are representative microscopic images (100× magnification (left panel)) and quantification of nuclear γH2AX-foci per cell (right panel). Data shown are the mean + SD of three independent experiments (*n* = 3). * *p* ≤ 0.05; Cis, cisplatin; Doxo, doxorubicin; IQ, 5-epi-ilimaquinone; NQ, 5-epi-nakijiquinone Q; SA, secalonic acid F; γH2AX, S139 phosphorylated H2AX; ns, not significant.

**Figure 6 molecules-27-03567-f006:**
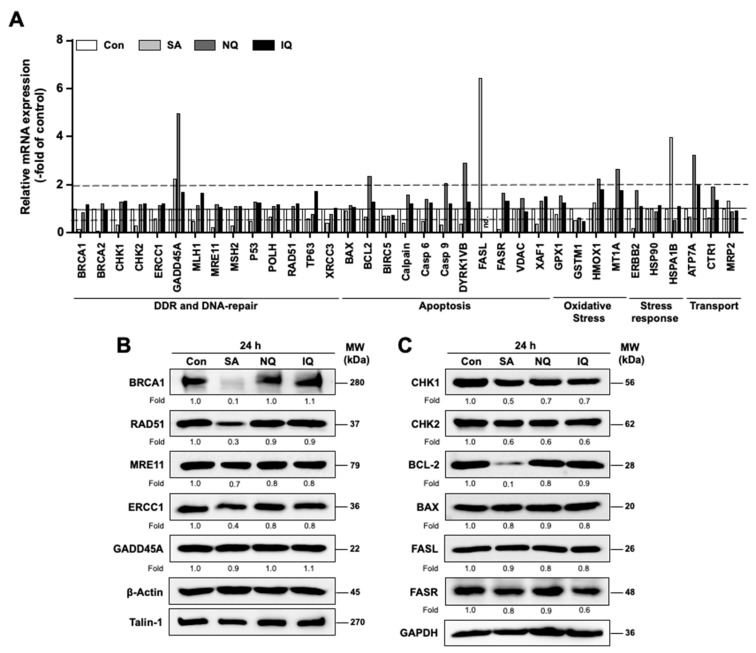
Influence of the NC on gene and protein expression of a subset of selected susceptibility-related factors. (**A**) mRNA expression of a selected subset of genes involved in DDR and DNA-repair, apoptosis, oxidative stress, stress response and transport were analyzed after 24 h mono-treatment of BxPC3 cells with SA (3 µM), NQ (2 µM) and IQ (8 µM) via quantitative RT-PCR. Data shown are the mean values from one representative experiment performed in triplicate. Pooled RNA samples isolated from three independent experiments (*n* = 3) were used for analysis. Relative mRNA expression in corresponding untreated cells (Con) was set to 1.0 (marked with continuous line). mRNA expression levels were normalized to β-actin. Only changes in mRNA expression of ≥2.0-fold and ≤0.5-fold were considered as biologically relevant (marked with dashed lines). Nd, not detectable. (**B**,**C**) Protein expression of repair- (**B**) and DDR-/apoptosis (**C**)-related factors was analyzed 24 h after mono-treatment of BxPC3 cells with SA, NQ and IQ. After incubation period of 24 h, BxPC3 cells were harvested for Western blot analyses. Relative protein expression in untreated cells, which were used as control (Con), was set to 1.0. Expression of β-actin, talin-1 or GAPDH was used as protein loading control. Shown are results of a representative experiment. BAX, BCL-2 associated X protein; BCL-2, B-cell lymphoma 2; BRCA1, BReast CAncer Gene 1; CHK1/2, checkpoint kinase 1/2; ERCC1, excision-repair cross-complementing 1; FASL, Fas ligand 1; FASR, Fas receptor; GADD45A, growth arrest and DNA damage-inducible alpha; Mre11, meiotic recombination gene 11; RAD51, RAD51 recombinase; IQ, 5-epi-ilimaquinone; NQ, 5-epi-nakijiquinone Q; SA, secalonic acid F.

**Figure 7 molecules-27-03567-f007:**
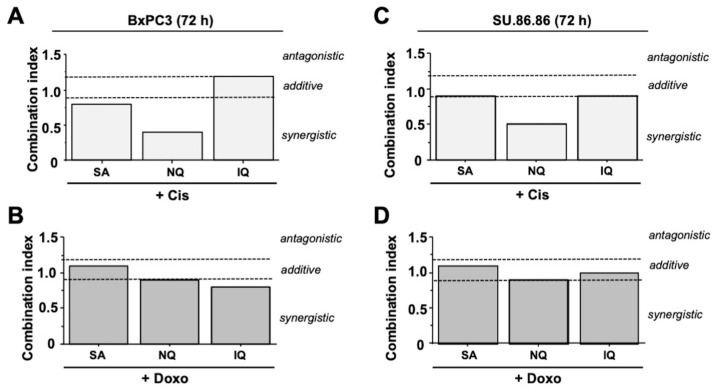
Influence of combination treatment on cell viability. In order to investigate the influence of the combination treatments on viability, BxPC3 (**A**,**B**) and SU.86.86 (**C**,**D**) cells were treated for 72 h with different concentrations of natural compounds and cisplatin (Cis) (**A**,**C**) or doxorubicin (Doxo) (**B**,**D**). Based on the respective IC_20_, IC_50_ and IC_80_ of the various compounds [42], the following concentrations of the individual substances were used for calculating the combination index (CI): Cis (1, 5 and 10 µM); Doxo (0.05, 0.3 and 1 µM); SA (2, 3 and 5 µM); NQ (1.5, 5 and 10 µM); IQ (0.5, 1.5 and 3.5 µM). Viability was determined by the Alamar Blue^®^ assay and the CI was calculated with the help of the CompuSyn software as described in methods. A distinction is made between antagonistic (CI > 1.2), additive (CI ≥ 0.8 ≤ 1.2) and synergistic (CI < 0.8) effects. IQ, 5-epi-ilimanquinone; NQ, 5-epi-nakijiquinone Q; SA, secalonic acid F. Detailed results of these extensive analyses are depicted in Appendix A.

**Figure 8 molecules-27-03567-f008:**
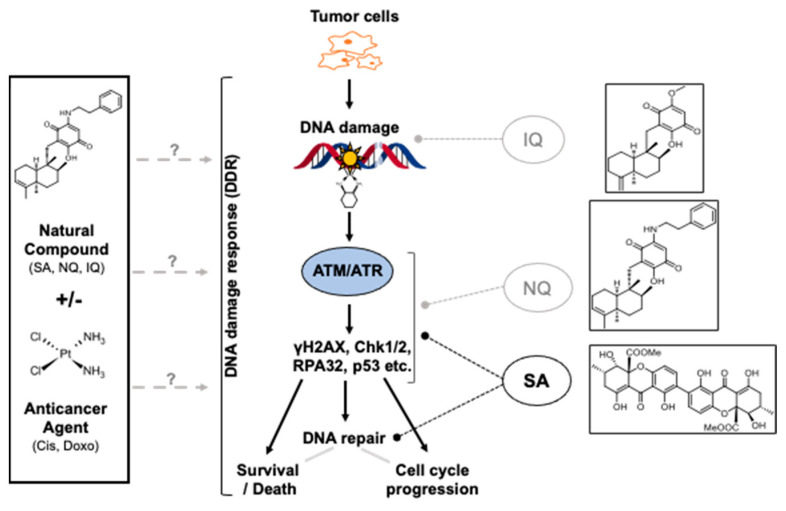
Model of the effects of the NC on pancreatic carcinoma cells. Based on the data available we hypothesize that the natural compounds investigated in this study (i.e., SA, NQ, IQ) can influence mechanisms of the DNA damage response (DDR). Most important, the NC also modulate the complex stress responses evoked by cAT. While IQ seems to affect DDR via formation of ROS (reactive oxygen species), NQ appears to influence the activity of DDR-related kinases. SA is suggested as the most promising NC because it interferes with the activity of several DDR related factors and represses the expression of genes coding for HR-related DSB repair factors. IQ, 5-epi-ilimanquinone; NQ, 5-epi-nakijiquinone Q; SA, secalonic acid F.

## Data Availability

Not applicable.

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
