# Peer review of "Targeting Mechanisms of the DNA Damage Response (DDR) and DNA Repair by Natural Compounds to Improve cAT-Triggered Tumor Cell Death"

_molecules, 2022, doi:10.3390/molecules27113567_

Round 1

Reviewer 1 Report

The study entitled “Targeting mechanisms of the DNA damage response (DDR)

and DNA repair by natural compounds to trigger tumor cell kill” investigated

the effects of natural compounds SA, NQ, and IQ on pancreatic cancer cells as a single agent as well as in combination with cisplatin and doxorubicin. In general, the manuscript is presented in a scattered manner. The rationale of various experiments, combination treatment, and chosen cell line model is missing. Maybe focusing on only one compound would make the manuscript more streamlined. Addressing the following points will further improve the manuscript.

Major points

  • The title of the study is misleading. The authors did not provide the mechanistic data on the compounds they evaluated. This manuscript simply described the consequence of exposure to SA, NQ, and IQ in two pancreatic cancer cells. It is unclear what is/are the molecular targets of these compounds and how they exert their anti-tumor activity.

  • The rationale to choose pancreatic cancer cells must be included.

  • For data provided in Figure 1, Western blot for total ATM, ATR, RPA32, KAP1, CHK1, CHK2, H2AX and p53 must be included.

  • Page 2, line 91: Authors stated that “Following co-treatment with Cis the protein levels of phosphorylated RPA32 and pKAP1 were further enhanced as compared to the corresponding mono-treatments while the protein level of pCHK1 was reduced (Figure 1)”. Authors must quantify the bolts and provide statistics to strengthen this claim.

  • Figure 3: Panel A and C, total caspase 3 and 7 blots must be included. Loading in Panel B does not seem convincing. For quantification, cleaved caspase 3 and 7 must be normalized to loading control of respective sample and then it should be compared to control.

  • Figure 5: In the text authors mentioned that they wanted to measure the kinetic of DSB repair, however they only provided data for one time point post-treatment (i.e., 24 h). Authors should include few more time points.

  • Figure 7: Survival curve for each individual treatment must be provided along with statistics.

  • Figure 7: Authors did not attempt to test the toxic effect of their compound on the normal pancreatic cell line. If they aim to develop these compounds as anti-tumor agents, authors must conduct some key experiments (survival and DDR induction) in a normal pancreatic cell line and show the selective toxicity against cancer cells.

Author Response

Reviewer #1

  1. The title of the study is misleading. The authors did not provide the mechanistic data on the compounds they evaluated. This manuscript simply described the consequence of exposure to SA, NQ, and IQ in two pancreatic cancer cells. It is unclear what is/are the molecular targets of these compounds and how they exert their anti-tumor activity.

It is correct that we have not yet identified the molecular target(s) of the three compounds. However, since target identification is highly challenging, it is not worthwhile before having extensive biological data at hand. Based on the results obtained from the mono-treatments with the three NC there is now already some information available regarding how they exert their toxicity, namely in a DNA damage dependent manner. However, and most important, the main aim of the study was to investigate the effects of the 3 NC if combined with cAT (i.e. Doxo and CisPt). We aimed to figure out whether and how the NC modify the cAT´s anticancer efficacy in combination treatment. Here, we have shown substantial mechanistical data (see below) and, therefore, do not agree with the reviewer´s criticism that mechanistic data were not presented. We have shown the following:

  1. a) In the context of our previous study [1] we already showed that the 3 NC under investigation do not affect Doxo uptake/efflux, indicating that their modulating effects on cAT-stimulated DDR is likely independent of drug transport.

  1. b) In the present study, we extended these data and summarized the results in the graphical abstract (Figure 8). Regarding SA, which we consider to be the most promising NC for forthcoming in vivo analyses, we demonstrate that this NC

- substantially modifies cAT-mediated activation (i.e. phosphorylation) of DDR-related factors involved in the regulation of replicative stress responses.

- inhibits cAT-mediated activation of G2/M cell cycle checkpoint control mechanisms and, furthermore, increases cAT-induced SubG1 fraction

- stimulates caspase 3/7 activity indicating that it promotes apoptotic cell death

- increases the steady state level of residual Doxo-induced nucelar gH2AX foci as well as CisPt-induced DNA intrastrand-crosslinks, indicating that it slows down DNA repair

- reduces the mRNA and protein expression of factors regulating DSB repair by homologous recombination (HR) (i.e. RAD51 and BRCA1)

Overall, these data strongly indicate that the antitumor activity of SA is related to its interference with DNA damage-related signaling mechanisms evoked by replicative stress, DNA damage repair (i.e. DSB repair), and abrogation of cAT-induced checkpoint control mechanisms, thereby eventually triggering caspase-3/7-mediated cell death. To our opinion this is relevant mechanistic information.

It goes without saying that a more detailed characterization of the molecular mechanisms, including primary target(s) identification, of three different compounds within the frame of one single manuscript is not possible. In currently ongoing and forthcoming studies we aim to identify the primary molecular target of SA as well as its in vivo efficacy and therapeutic window.

  1. The rationale to choose pancreatic cancer cells must be included.

This is a good point, indeed. The majority of the molecular analysis was performed in pancreatic carcinoma cells expressing wild-type Ki-Ras (i.e. BxPC3) or mutated Ki-Ras (i.e. SU8686), because this model was already used in our previous study [1], having in mind the poor prognosis of patients suffering from pancreatic cancer. So, when extending our studies on the 3 pre-selected NS, we decided to retain the same model system. This is clarified in the revision now.

In this context, we would like to emphasize that we included several additional tumor cell lines in our study. Cytotoxicity data obtained with these tumor cell lines are presented in Supplementary Fig. 5 of the manuscript.

  1. For data provided in Figure 1, Western blot for total ATM, ATR, RPA32, KAP1, CHK1, CHK2, H2AX and p53 must be included.

Regarding this point, it is important to have in mind that it is the phosphorylated form of the analyzed DDR factors that is the biochemically active form. In consequence, it is only the phosphorylated protein that is functionally relevant. Therefore, from our point of view, it is not mandatory to additionally monitor the protein levels of all non-phosphorylated forms. Moreover, in the shortness of time that is available for resubmission of a revised manuscript, it is simply not possible to repeat all of these extensive Western blot analyses. Nevertheless, in view of the reviewer´s comment, we analyzed the non-phosphorylated levels of some selected DDR-related factors (i.e. CHK1 and P53) and included these data into Figure 1 of the revised manuscript.

  1. Page 2, line 91: Authors stated that “Following co-treatment with Cis the protein levels of phosphorylated RPA32 and pKAP1 were further enhanced as compared to the corresponding mono-treatments while the protein level of pCHK1 was reduced (Figure 1)”. Authors must quantify the bolts and provide statistics to strengthen this claim.

The blots were quantified as suggested and data were included into Figure 1 now. In the legends of Figure 1 we clarified that the blots shown are representative data (of two experiments). Moreover, it is important to note that we only consider an increase in protein expression of more than 50 % as potentially meaningful. So, only NC-mediated alterations in cAT-stimulated phosphorylation levels of DDR-related proteins >1.5-fold (or < 0.7-fold) as compared to the corresponding cAT-treated are considered as biologically noticable. This is clarified in the legends of the revised manuscript now.

  1. Figure 3: Panel A and C, total caspase 3 and 7 blots must be included. Loading in Panel B does not seem convincing. For quantification, cleaved caspase 3 and 7 must be normalized to loading control of respective sample and then it should be compared to control.

It is well established that - upon cleavage of the corresponding pro-caspase - it is the cleaved caspase that is the active and, therefore, the biologically relevant form. Hence, to our opinion, measuring pro-caspase 3/7 protein levels is not mandatory as it does not provide any additional information. Nevertheless, following the reviewer´s recommendation, we included data for pro-caspase 3 now and, moreover, added data showing PAPR cleavage after 24 h treatment with SA. In this context we would like to emphasize that we not only measured cleavage of pro-caspases or PARP but analyzed caspase-3/7 activity, too. These data are presented in Figure 3B and 3D and support the conclusion that SA and NQ influence caspase-3/7-mediated apoptosis in a cAT-specific manner.

  1. Figure 5: In the text authors mentioned that they wanted to measure the kinetic of DSB repair, however they only provided data for one time point post-treatment (i.e., 24 h). Authors should include few more time points.

We agree with the reviewer that we did not measure DSB repair kinetics. Rather, we monitored the steady state level of residual DNA damage (i.e. DSB) as measured 24 h after cAT pulse-treatment. This is clarified in the revised manuscript.

  1. Figure 7: Survival curve for each individual treatment must be provided along with statistics.

The original data of these highly extensive analyses are presented in the Supplementary Figure 4 of the revised manuscript now.

  1. Figure 7: Authors did not attempt to test the toxic effect of their compound on the normal pancreatic cell line. If they aim to develop these compounds as anti-tumor agents, authors must conduct some key experiments (survival and DDR induction) in a normal pancreatic cell line and show the selective toxicity against cancer cells.

Indeed, in view of possible adverse effects of the NC, their cytotoxic potency evoked on normal cells should be analyzed. Yet, from a toxicological point of view, such analyses do not necessarily need to be performed by use of normal pancreatic cells. This is because the cell type-specificity of an adverse drug effect is poorly predictable. Noteworthy in this context, we have already analyzed the cytotoxicity of the 3 NC employing non-malignant primary human fibroblast cells as a normal cell model. Comparing the IC50 of the various tumor cell lines (Supplementary Figure S5) with that of the human fibroblasts (Supplementary Figure S6) we assume an unfavorable therapeutic window for NQ, while SA seems to be much better tolerated by non-malignant cells.

For a more detailed and toxicologically meaningful assessment of normal tissue damage, in vivo studies aiming to analyse adverse effects in different types of tissue are mandatory. Such experiments will be part of our forthcoming in vivo studies where we will investigate the anti-tumor potency of SA. In this study we will monitor the therapeutic window of SA in mono- or combined treatment regimen by measuring different tissue toxicities. Having in mind the clinically relevant dose limiting adverse effects of CisPt and Doxo, we will focus on the analysis of hemato-, hepato- and nephrotoxicity as well as toxic effects on the heart. In view of the reviewer´s comment, we now plan to include the pancreas in our analyses as well.

Reviewer 2 Report

Targeting mechanisms of the DNA damage response (DDR)  and DNA repair by natural compounds to trigger tumor cell kill. Overall the paper is good some modification need to be done

  1. Please modify the tile kill can be changed to death
  2. Authors can give some recent literature of natural compounds as a source of anticancer drugs authors may get benefitted by  https://doi.org/10.1016/B978-0-323-85503-7.00033-X
  3. Similiarity index is some what more please reduce.
  4. mention the ic50 of the natural compounds in the abstract.
  5. check for spelling and grammatical errors 

Author Response

Reviewer #2

  1. Please modify the tile kill can be changed to death

The title of the manuscript was modified as suggested.

  1. Authors can give some recent literature of natural compounds as a source of anticancer drugs authors may get benefitted by  https://doi.org/10.1016/B978-0-323-85503-7.00033-X

We thank the reviewer for this suggestion and included more topical literature regarding natural compounds as an important source of anticancer drugs.

  1. Similiarity index is some what more please reduce.

We do apologize but we do not understand this comment of the reviewer. Does the reviewer refer to the combination index (CI) here?

  1. Mention the ic50 of the natural compounds in the abstract.

The major intention of our manuscript was to investigate the effects of the 3 NC if combined with the cAT Doxo or CisPt. IC50 of the NC on pancreatic carcinoma cells if used as a mono-treatment were already reported before [1] and were added to Supplementary Figure S5 for reason of comparison.

  1. Check for spelling and grammatical errors 

The manuscript was carefully revised to correct spelling and grammatical errors.

Round 2

Reviewer 1 Report

In the revised manuscript, the authors made some changes but did not address several of my previous comments. For example, it is a standard practice in most fields including the DNA repair field that the analysis of phosphorylated protein via Western blotting must be accompanied by total protein level. I disagree with most of the explanations provided by the authors to address my previous comments.

Author Response

Reviewer #1

  1. The title of the study is misleading. The authors did not provide the mechanistic data on the compounds they evaluated. This manuscript simply described the consequence of exposure to SA, NQ, and IQ in two pancreatic cancer cells. It is unclear what is/are the molecular targets of these compounds and how they exert their anti-tumor activity.

It is correct that we have not yet identified the molecular target(s) of the three compounds. However, since target identification is highly challenging, it is not possible to identify three novel targets within one single study. Moreover, to our opinion, it is not worthwhile the complex and extensive work that is required for target identification before having more biological data at hand. Based on the results obtained in the present study from the mono-treatments with the three NC, there is novel and noticeable information available now how the NC, especially SA, cause cytotoxicity. The data indicate that SA is interfering with replicative stress responses, DNA repair, checkpoint control mechanisms and caspase mediate apoptosis.

However, and most important to note, it was the main aim of our study to investigate the modulatory effects of the 3 NC on DDR-related stress responses if combined with conventional anticancer drugs (cAT) (i.e. Doxo and Cis). This is clarified in the revision. Having this in mind we changed the title of our manuscript accordingly. In this regard, we did provide substantial mechanistical data (see below) because we have shown the following:

  1. a) In the context of our previous study [1] we already showed that the 3 NC under investigation do not affect Doxo uptake/efflux, indicating that their modulating effects on cAT-stimulated DDR is likely independent of drug transport. This is additionally mentioned now in the text.

  1. b) In the present study, we extended these data and summarized the results in the graphical abstract (Figure 8). Regarding SA, which we consider to be the most promising NC for forthcoming in vivo analyses, we demonstrate that this NC

- substantially modifies cAT-mediated activation (i.e. phosphorylation) of DDR-related factors that are well known to be involved in the regulation of replicative stress responses.

- inhibits cAT-mediated activation of G2/M cell cycle checkpoint control mechanisms and, furthermore, increases cAT-induced SubG1 fraction

- stimulates caspase 3/7 activity indicating that it promotes apoptotic cell death

- increases the steady state level of residual Doxo-induced nucelar gH2AX foci as well as CisPt-induced DNA intrastrand-crosslinks, indicating that it slows down DNA repair

- reduces the mRNA and protein expression of factors that are well-known to regulate DSB repair by homologous recombination (HR) (i.e. RAD51 and BRCA1)

Overall, these data strongly indicate that the antitumor activity of SA is related to its interference with DNA damage-related signaling mechanisms evoked by replicative stress and DNA damage repair (i.e. DSB repair) leading to dysregulation of cAT-induced checkpoint control mechanisms, thereby eventually triggering caspase-3/7-mediated cell death. This is clarified in the revised manuscript now. To our opinion this is relevant and novel mechanistic information worthwhile to be published in Molecules.

A more detailed characterization of the molecular mechanisms, most important the primary target(s) identification of SA, as well as its anticancer efficacy and therapeutic window in vivo is subject of currently ongoing and forthcoming studies. This is also emphasized now in the revision.

  1. The rationale to choose pancreatic cancer cells must be included.

This is a good point, indeed. The majority of the molecular analysis was performed in pancreatic carcinoma cells expressing wild-type Ki-Ras (i.e. BxPC3) or mutated Ki-Ras (i.e. SU8686), because this model was already used in our previous study [1], having in mind the poor prognosis of patients suffering from pancreatic cancer. So, when extending our studies on the 3 pre-selected NS, we decided to retain the same model system. This is clarified in the revision now.

In this context, we would like to emphasize that we included several additional tumor cell lines in our study. Cytotoxicity data obtained with these tumor cell lines are presented in Supplementary Fig. 5 of the manuscript.

  1. For data provided in Figure 1, Western blot for total ATM, ATR, RPA32, KAP1, CHK1, CHK2, H2AX and p53 must be included.

Regarding this point, it is important to have in mind that it is the phosphorylated form of the analyzed DDR factors that is the biochemically active form. In consequence, it is only the phosphorylated protein that is functionally relevant. Therefore, from our point of view, it is not mandatory to additionally monitor the protein levels of all non-phosphorylated forms. Moreover, in the shortness of time that is available for resubmission of a revised manuscript, it is simply not possible to repeat all of these extensive Western blot analyses. Nevertheless, in view of the reviewer´s comment, we analyzed the non-phosphorylated levels of some selected DDR-related factors (i.e. CHK1 and P53) and included these data into Figure 1 of the revised manuscript.

  1. Page 2, line 91: Authors stated that “Following co-treatment with Cis the protein levels of phosphorylated RPA32 and pKAP1 were further enhanced as compared to the corresponding mono-treatments while the protein level of pCHK1 was reduced (Figure 1)”. Authors must quantify the bolts and provide statistics to strengthen this claim.

The blots were quantified as suggested and data were included into Figure 1 now. In the legends of Figure 1 we clarified that the blots shown are representative data (of two experiments). Moreover, it is important to note that we only consider an increase in protein expression of more than 50 % as potentially meaningful. So, only NC-mediated alterations in cAT-stimulated phosphorylation levels of DDR-related proteins >1.5-fold (or < 0.7-fold) as compared to the corresponding cAT-treated are considered as biologically noticable. This is clarified in the legends of the revised manuscript now.

  1. Figure 3: Panel A and C, total caspase 3 and 7 blots must be included. Loading in Panel B does not seem convincing. For quantification, cleaved caspase 3 and 7 must be normalized to loading control of respective sample and then it should be compared to control.

It is well established that - upon cleavage of the corresponding pro-caspase - it is the cleaved caspase that is the active and, therefore, the biologically relevant form. Hence, to our opinion, measuring pro-caspase 3/7 protein levels is not mandatory as it does not provide any additional information. Nevertheless, following the reviewer´s recommendation, we included data for pro-caspase 3 now and, moreover, added data showing PAPR cleavage after 24 h treatment with SA. In this context we would like to emphasize that we not only measured cleavage of pro-caspases or PARP but analyzed caspase-3/7 activity, too. These data are presented in Figure 3B and 3D and support the conclusion that SA and NQ influence caspase-3/7-mediated apoptosis in a cAT-specific manner.

  1. Figure 5: In the text authors mentioned that they wanted to measure the kinetic of DSB repair, however they only provided data for one time point post-treatment (i.e., 24 h). Authors should include few more time points.

We agree with the reviewer that we did not measure DSB repair kinetics. Rather, we monitored the steady state level of residual DNA damage (i.e. DSB) as measured 24 h after cAT pulse-treatment. This is clarified in the revised manuscript.

  1. Figure 7: Survival curve for each individual treatment must be provided along with statistics.

The original data of these highly extensive analyses are presented in the Supplementary Figure 4 of the revised manuscript now.

  1. Figure 7: Authors did not attempt to test the toxic effect of their compound on the normal pancreatic cell line. If they aim to develop these compounds as anti-tumor agents, authors must conduct some key experiments (survival and DDR induction) in a normal pancreatic cell line and show the selective toxicity against cancer cells.

Indeed, in view of possible adverse effects of the NC, their cytotoxic potency evoked on normal cells should be analyzed. Yet, from a toxicological point of view, such analyses do not necessarily need to be performed by use of normal pancreatic cells. This is because the cell type-specificity of an adverse drug effect is poorly predictable. Noteworthy in this context, we have already analyzed the cytotoxicity of the 3 NC employing non-malignant primary human fibroblast cells as a normal cell model. Comparing the IC50 of the various tumor cell lines (Supplementary Figure S5) with that of the human fibroblasts (Supplementary Figure S6) we assume an unfavorable therapeutic window for NQ, while SA seems to be much better tolerated by non-malignant cells. This is discussed in the revised manuscript now.

For a more detailed and toxicologically meaningful assessment of normal tissue damage, in vivo studies aiming to analyse adverse effects in different types of tissue are mandatory. Such experiments will be part of our forthcoming in vivo studies where we will investigate the anti-tumor potency of SA. In this study we will monitor the therapeutic window of SA in mono- or combined treatment regimen by measuring different tissue toxicities. Having in mind the clinically relevant dose limiting adverse effects of CisPt and Doxo, we will focus on the analysis of hemato-, hepato- and nephrotoxicity as well as toxic effects on the heart. In view of the reviewer´s comment, we now plan to include the pancreas in our analyses as well.

We hope that we have dealt satisfactorily with the major comments/criticisms of the reviewer and that or thoroughly revised manuscript is acceptable now for publication in the above-mentioned special issue of Molecules.
